# The Effect of Ginger Rhizome Addition and Storage Time on the Quality of Pork Meatloaf

**DOI:** 10.3390/foods11223563

**Published:** 2022-11-09

**Authors:** Mirosława Karpińska-Tymoszczyk, Anna Draszanowska, Marzena Danowska-Oziewicz

**Affiliations:** Department of Human Nutrition, University of Warmia and Mazury in Olsztyn, Słoneczna 45 f, 10-718 Olsztyn, Poland

**Keywords:** pork meatloaf, ginger rhizome, natural antioxidant, storage, physicochemical properties, sensory quality

## Abstract

This study investigated the effect of ginger-rhizome addition and storage time on the physicochemical and sensory quality of pork meatloaf. Three types of pork meatloaf were evaluated: control and with 1% and 2% addition of ginger. All meatloaves were vacuum packaged and stored for 0, 7, 14, and 21 days at 4 °C. The addition of ginger rhizome significantly reduced lipid oxidation, and the higher inclusion rate was more effective in this regard. Ginger decreased red-colour saturation (a*) and increased colour brightness. The addition of ginger rhizome at 2% induced a greater decrease in meat hardness and improved chewiness in comparison with 1% addition. Products containing ginger differed from the control sample in aroma, texture, and taste, but no significant differences were found in the overall quality of the compared samples.

## 1. Introduction

Lipid oxidation is the main non-microbial cause of quality deterioration in meat and meat products. This process leads to the loss of unsaturated fatty acids and vitamins, and it decreases the nutritional value of meat and meat products. The first undesirable change induced by oxidation is the deterioration in the sensory quality of meat. Oxidation also affects meat colour and texture, and it produces rancid odour and flavour which affect consumer acceptance [1]. Natural and synthetic antioxidants are added to meat products to prevent lipid oxidation, retard the development of off-flavours, and improve colour stability [2]. Due to the proven negative effects of synthetic additives on human health, alternative additives of natural origin are being sought to protect food from spoilage and increase consumer safety [3]. Natural antioxidants are regarded as ingredients that are obtained from natural sources and exhibit antioxidant properties in food model systems. Natural antioxidants play a very important role in the food industry and are presently used more often than synthetic antioxidants [2]. In addition, modern consumers have a preference for natural products, which encourages meat producers to limit the use of synthetic additives and resort to natural additives with similar properties. Consumers are also increasingly health-conscious, which increases the demand for functional foods around the world. Therefore, the interest in new natural additives that improve the stability and safety of food is a valid research concern [4].

Ginger has a high nutritional value and is a rich source of starch, cellulose, protein, amino acids, and trace minerals (copper, iron, manganese, zinc, chromium, nickel, cobalt). Ginger contains 40–60% of starch, 6.2–19.8% of protein, 5.7–14.5% of total lipids, and 1.1–7.0% of cellulose on a dry-matter basis. Raw ginger contains active ingredients such as gingerol, 6-shogaol, curcumin, and ginger protease. These ingredients are responsible for ginger’s functional properties, including antioxidant, anticarcinogenic, and anti-inflammatory properties [5]. The antioxidant activity of ginger can be attributed mainly to the presence of gingerols, shoals, and zingerone [6].

Draszanowska et al. [7] demonstrated that the addition of ginger rhizome was as effective as sodium ascorbate in inhibiting oxidative changes during refrigerated storage of pasteurised canned meats. Mancini et al. [8] reported that the addition of ginger powder to rabbit feed inhibited oxidative processes during refrigerated storage of raw rabbit meat. Putra et al. [9] found that marinating goat meat in fresh ginger limited lipid oxidation in raw and cooked meat during refrigerated storage.

Pork products are susceptible to oxidation processes during storage, which can degrade their quality and therefore they require the use of additives to reduce these processes. A current trend in food production is the search for natural additives that can replace synthetic ones. Therefore, the purpose of this research was to test ginger rhizome as an additive to pork meatloaf and to evaluate its effects in terms of oxidative changes during storage by instrumentally measuring parameters of colour and texture as well as by assessing the sensory quality of products.

## 2. Materials and Methods

### 2.1. Materials

Fresh ginger (Zingiber officinale, Chinese yellow ginger) was purchased from a local market. Pork shoulder meat was acquired from a pig farm in the region of Warmia and Mazury (Poland). Pork was obtained from Polish landrace pigs aged 5–6 months, weighing 115 kg, 24 h after slaughter.

### 2.2. Sample Preparation

Ginger rhizomes were peeled, washed, and grated on a grater with a mesh diameter of 2 mm. Shoulder meat was cleaned, washed, dried, cut into 40 mm × 40 mm × 40 mm pieces, and ground in a Mesko-AGD KU2-4E machine (Skarżysko Kamienna, Poland) using a 5 mm mesh plate. Pork meatloaves consisted of pork shoulder (90%), liquid eggs (5%), and water (5%). The other ingredients: salt (1%) and ginger rhizome (1% and 2%) were added in relation to total mass weight. Ground meat was mixed in a Mankiewicz SP-100A-B mixer (Radzionków, Poland) for 10 min to homogenise the mass. Homogenised meat was divided into three portions. The first portion was combined with liquid eggs and salt (control sample); the second portion was combined with liquid eggs, salt, and 1% ginger (sample G1); and the third portion was combined with liquid eggs, salt, and 2% ginger (sample G2). Salt and ginger were first combined with a small amount of each meat portion; the remaining meat was then added, and each portion was mixed for 15 min in a Mankiewicz SP-100A-B mixer (Radzionków, Poland). From the prepared portions, 250 g was weighed, placed in an aluminium baking tin (40 subsamples of each treatment), and roasted in a Rational SCCWE-101 steam–convection oven (Landsberg am Lech, Germany) equipped with a measuring probe at 180 °C until the temperature inside meat samples reached 72 °C. Baked samples were left to stand at room temperature for 30 min, after which they were cooled to storage temperature (4 °C) in a Bartscher AL5 BT700605 blast freezer (Salzkotten, Germany). Cooled meatloaves were vacuum-packaged in bags made of 52 μm thick PA/PE multilayer barrier film (Hendi, Lamprechtshausen, Austria) using an Edesa VAC-20 DT chamber vacuum sealer (Barcelona, Spain).

Pork meatloaves were stored at a constant temperature (4 °C) in a MediLine Lkexv 3600 laboratory refrigerator (Liebherr, Austria) for 24 h (time 0) and 7, 14, and 21 days. Ten pork meatloaves from each treatment were randomly selected in each stage of the study for analyses. Of these, three meatloaves were used in physicochemical analyses, two in colour analyses, two in texture analyses, and two in sensory evaluations. The experiment was performed in triplicate.

### 2.3. Preparation of Samples for Physical and Chemical Analysis

At each stage of the experiment, three pork meatloaves from each treatment were randomly selected and ground separately in a Zelmer ZMM10891 mincing machine (Rzeszów, Poland) with a 3 mm grinder plate. The ground samples were mixed and placed in a glass container. The prepared samples were used in chemical composition analyses and measurements of the TBARS (thiobarbituric-acid reactive substances) index, pH, and water activity.

### 2.4. Proximate Composition

Protein content was determined by the Kjeldahl method [10], fat content was determined by the Soxhlet method [11], and water content was determined by drying samples to constant weight at 105 °C [12].

### 2.5. pH Values

A 10 g sample was weighed, combined with 10 mL of distilled water, and homogenised in an Edmund Bűhler GmbH HO 4A homogeniser (Hechingen, Germany) for 3 min at 6000 rpm. The pH was measured using a Hanna Instrument 210 pH meter (Woonsocket, RI, USA) at room temperature. Prior to measurement, the instrument was calibrated using buffers with pH 7 and pH 4.

### 2.6. Water Activity

Three samples selected randomly from each treatment were homogenised separately and placed in the sample chamber of the AWC 2000 analyser (Novasina, Pfäffikon, Switzerland) to measure water activity. The apparatus was calibrated at 20 °C before measurement. Each sample was measured in triplicate, and the results were averaged.

### 2.7. Lipid Oxidation (TBARS)

Oxidative changes were determined by calculating the TBARS index according to the method of Salih et al. [13]. For this purpose, a 10 g sample was weighed and homogenised with 34.35 mL of chilled (4 °C) 4% perchloric acid and 0.75 mL of 0.01% alcoholic BHT solution in the Edmund Bühler GmbH HO 4A (Hechingen, Germany) homogeniser for 2 min at 4000 rpm. Subsequently, the homogenate was filtered through Whatman 1 blotting paper into 50 mL measuring cylinders. The resulting filtrate was made up to 50 mL by washing the precipitate formed on the filter with perchloric acid. The filtrate was stirred, and specimens of 5 mL were transferred to 20 mL test tubes. Five milliliters of 0.02 M aqueous solution of 2-thiobarbituric acid was added. The test tubes were capped and heated in a boiling water bath for 1 h. The tubes were then cooled for 10 min under cold running water. The prepared samples were used to measure absorbance at a wavelength of 532 nm with the Optizen POP UV/VIS spectrophotometer (Metasys Co. Ltd., Deajeon, South Korea) against a blank sample containing 5 mL of 4% perchloric acid and 5 mL of the TBA reagent. Three measurements were performed for each formulation. The TBARS value was calculated with the use of the below formula and expressed in mg of malondialdehyde (MDA) per 1 kg of the product [14]:TBARS = A × K (mg MDA/kg)(1)
where A is the absorbance of the analysed sample and K is the conversion factor of 5.5.

### 2.8. Colour Parameters

Colour was analysed in the CIE Lab system (L*, a*, b*) using a Konica Minolta CR-400 chromameter (Osaka, Japan) with a measurement area of 8 mm. The chromameter was calibrated before measurement using a white standard plate with Y = 89.3, x = 0.3159, and y = 0.3225. The measurements were performed using illuminant D65 and standard observer 2. The colour saturation (C*) and huge angle (h°) were calculated according to formulae:(2)$$C^{*}=\sqrt{a^{*2}+b^{*2}}$$
(3)$$h{}^{\circ}\;=\arctan\;\left(\frac{b^{*}}{a^{*}}\right)$$
where $C^{*}\;\mathrm{is}\;\mathrm{the}\;$colour saturation, h° is the huge angle, a* is the redness, and b* is the yellowness.

Two meatloaves were selected randomly, and three slices with a thickness of 12 mm each were cut from each samples. Colour measurements were performed at three randomly selected points on the surface of each slice, and then the results were averaged.

### 2.9. Texture Profile Analysis

Six cubes measuring 10 × 10 × 10 mm each were cut from three randomly selected meatloaves of each type. The measurements were performed at room temperature using a TA.XT. plus texture analyser (Stable Micro System, Goldalming, UK) equipped with a 50 kg load cell. The samples were double compressed to 50% of the original height. The speed of the compression element during the test was 5 mm/s. The results were recorded with the use of the Texture Expert version 1.22 software.

### 2.10. Sensory Evaluation

The sensory evaluation of the pork meatloaves was performed by 10 panellists (8 women and 2 men; aged 21 to 57). The panellists were students and employees of the Faculty of Food Science at the University of Warmia and Mazury in Olsztyn (Poland). The panellists were trained in sensory analysis according to Standard ISO 8586 [15] and the applied evaluation method. The samples were evaluated in a sensory-analysis laboratory. Two randomly selected pork meatloaves of each type were used for sensory evaluation. Before preparing the samples for evaluation, they were warmed to room temperature. The meatloaves were cut into slices with a thickness of 5–6 mm each, and two slices (one of each sample) were served to the evaluators on white porcelain plates covered with aluminium foil coded with a random three-digit code. Water was served to the panellists to cleanse the palate between samples. The samples were assessed on a numerical-interval scale ranging from 1 to 10 points [16]. The following distinguishing features were evaluated: intensity of aroma (roasted meat and ginger), texture (juiciness, cohesiveness, and tenderness), intensity of flavour (roasted meat and ginger), and overall quality (Table 1).

### 2.11. Statistical Analysis

The results were analysed statistically using Statistica version 13.3 software (TIBCO Software Inc., Tulsa, OK, USA). Mean values and standard deviation were calculated. The effects of experimental factors (additive and storage time) were determined by two-way analysis of variance (ANOVA). The significance of differences between means was checked with Tukey’s post hoc test at a significance level of *p* < 0.05.

## 3. Results and Discussion

### 3.1. Proximate Composition, pH, and Water Activity

Moisture content was determined as 70.56% in the control product. It was significantly lower (70.17%) in the sample with 1% ginger addition, and similar to the control sample in the meatloaf with 2% ginger addition (70.78%). The protein content of the evaluated samples ranged from 21.96% to 22.45%, and fat content ranged from 6.66% to 6.87%. The addition of ginger had no effect on the protein or fat content of the analysed products (Table 2).

The pH of the control sample ranged from 5.98 to 6.14 (Table 2) The addition of ginger significantly increased this parameter, and only after 14 days of storage the sample with 2% ginger addition had a significantly higher pH than that containing 1% ginger. Teref [17] found that the addition of ginger powder at 1%, 3%, and 5% to ground red meat significantly decreased pH values, and higher ginger inclusion rates increased the observed reduction in meat pH. No significant changes in pH were observed during storage in the meatloaves with 1% ginger addition. In the control sample, pH decreased significantly only after 14 days of storage, and it increased significantly after 21 days of storage. In the meatloaves with 2% ginger addition, pH continued to increase throughout the storage period (from 6.14 to 6.22). Olatidoye et al. [18] reported a steady increase in the pH of ground beef containing ginger extract during 8 days of refrigerated storage. However, samples with ginger extract were characterised by significantly lower pH than samples without this additive. The increase in pH during storage could be attributed to microbial growth and the breakdown of product ingredients [19].

Ginger addition and storage time had no effect on water activity (Table 2). This parameter was relatively high, in the range of 0.975 to 0.977, which could favour microbial growth and, consequently, a decrease of the storage life of samples. Water activity is an important indicator of the state of water in food. It largely determines microbial, chemical, and biochemical stability, as well as the physical properties of food products [20]. This parameter influences food safety and should be monitored to minimise health risks for consumers because microorganisms grow best within an aw range of 0.995–0.980, while most microorganisms stop growing at aw <0.900 [21].

### 3.2. Lipid Oxidation

Thiobarbituric acid (TBA) is commonly used as an indicator of lipid oxidation, and in the second oxidation step, peroxides are oxidised to aldehydes and ketones and TBARS are produced [21].

The effects of ginger addition and refrigerated storage time on oxidative changes in the pork meatloaves are shown in Table 2. Ginger addition significantly (*p* < 0.05) influenced the lipid oxidation index (TBARS) in the analysed samples. Products with both 1% and 2% ginger-rhizome addition contained significantly less malondialdehyde (MDA) than the control sample. Significant differences between samples with 1% and 2% ginger-rhizome addition were observed in MDA content after 0 and 21 days of refrigerated storage, and MDA levels were significantly lower in meatloaf containing 2% ginger.

Draszanowska et al. [7] reported that 1.5% ginger-rhizome addition to pasteurised canned pork meat was as effective as sodium ascorbate in inhibiting oxidative changes during refrigerated 50 days storage. Singh et al. [22] observed that 3% addition of ginger paste to chicken meat emulsion was as effective as 2% addition of garlic paste, but less effective than 0.2% addition of clove powder in inhibiting oxidative changes during refrigerated storage. Stoilova et al. [23] found that the antioxidant activity of ginger extract was similar to that of BHT. Abdel-Naeem and Mohamed [24] added ginger extract powder (7%) to camel-meat burgers and found that the additive improved lipid stability after 3 months of frozen storage (−18 °C). Lower TBARS values in ginger-enriched samples could be related to the activity of peroxide-scavenging enzymes, which inhibit the oxidation of unsaturated fatty acids [21].

In the present study, refrigerated storage time significantly (*p* < 0.05) affected the MDA content of the analysed meatloaves and ambiguous oxidative changes were noted in all samples during storage. The greatest fluctuations in MDA content were found in the control sample. In samples with the addition of ginger, MDA levels continued to increase until storage day 14, and decreased significantly after 21 days of storage.

### 3.3. Colour Parameters

The colour parameters of the analysed products are presented in Table 3. Colour parameter L* was significantly influenced only by 2% ginger addition, and the sample containing 2% ginger was significantly lighter than the control sample after 14 and 21 days of storage. The meatloaves with 1% ginger addition was characterised by similar values of L* throughout storage, whereas a steady increase in this parameter was observed in the control sample and in the sample with 2% ginger addition. The higher L* values of meatloaf with 2% ginger addition than control samples presumably may be attribute to the effect of ginger components on the pigment of this samples [24].

Ginger-enriched samples were characterised by lower redness (a*) values than the control sample, but not all differences were significant. During storage, redness values of the control sample and the sample with 2% ginger addition remained stable for 14 days. In the product with 1% ginger addition, the value of a* increased from 7.23 to 8.62 in the first 7 days of storage and remained unchanged until the end of the experiment (Table 2).

The addition of ginger had no significant effect on the values of yellowness (b*) in the analysed pork meatloaves. In the sample with 1% ginger addition, this colour parameter remained stable throughout storage. In the control sample, a significant reduction in b* value was noted in the last stage of the study. In the sample with 2% ginger addition, b* value decreased significantly after 7 days of storage and remained stable until the end of the experiment.

Colour saturation (C*) was similar on storage days 0 and 14, whereas the sample with 2% ginger addition was characterised by significantly lower C* values than the control sample on storage days 7 and 21. Storage time had no significant effect on colour saturation in the analysed pork meatloaves.

The addition of ginger induced a significant shift from red to orange and significantly increased hue (h°) value. During storage, significant changes in h° values were observed in the control sample and in the sample with 1% ginger addition. In the control sample, a significant change in the colour angle and a shift from red to orange were noted in the last stage of storage. In contrast, in the sample with 1% ginger addition, a shift from orange to red was observed after the first 7 days of storage.

Singh et al. [22] noted that 3% addition of ginger paste had no significant influence on colour parameters L*, a*, and b*; saturation; or colour angle in raw-chicken-meat emulsion. In contrast, Abdel-Naeem et al. [24] reported significantly higher values of parameter L* in burgers with 7% addition of ginger extract than in the control sample, whereas parameters a* and b* were similar in the experimental and control samples. Frank et al. [21] found that the addition of powdered and fresh ginger to cover brine in which silver carp were immersed before heat treatment significantly decreased the values of colour parameters L*, a*, and b* compared with the control sample (without the addition of ginger).

### 3.4. Texture-Profile Analysis

Ginger addition and storage time significantly affected the texture parameters of the tested pork meatloaves (Table 4). Samples with ginger addition were softer than the control one, and the sample with 2% ginger addition was characterised by significantly lower hardness values than the control sample on all storage days. The sample with 1% ginger addition differed significantly from the control sample on storage days 0 and 21. The hardness of the control product and the sample with 1% ginger addition decreased with storage time, whereas the hardness of the sample with 2% ginger addition remained similar throughout the entire storage period.

The springiness of the control meatloaf ranged from 0.81 to 0.82. In the sample with 1% ginger addition, springiness was significantly lower only on storage day 14. Ginger added at 2% significantly reduced springiness values to 0.72–0.77. During storage, significant changes in springiness were noted in samples with ginger. In the sample containing 1% ginger, springiness decreased after 14 days and increased significantly after 21 days of storage. In the sample with 2% ginger addition, springiness remained stable until day 14 and increased significantly after 21 days of storage.

Ginger added at 1% exerted a minor effect on the cohesiveness of the examined products. On most storage days, cohesiveness values were similar in the sample with 1% ginger addition and the control sample, and they were significantly lower in the sample with 1% ginger addition than in the control sample only on storage day 14. Cohesiveness was significantly lower in the sample with 2% ginger addition throughout the entire storage period. Storage time had no effect on the cohesiveness of the analysed meatloaves.

The addition of ginger at both 1% and 2% significantly improved the chewiness of meatloaves. Storage time had a significant effect on chewiness only in ginger-enriched samples, and chewiness decreased significantly after 14 days of storage.

In summary, the addition of ginger to pork meatloaf reduced hardness and springiness, and improved chewiness. These results are consistent with the findings of Draszanowska et al. [7] who analysed pasteurised canned meat with 1.5% addition of ginger rhizome.

Abdeldaiem and Ali [25] found that the addition of ginger extract to camel meat cuts (15%, 30%, and 45%) significantly reduced shear force and improved meat tenderness. Similar observations were made by Naveena and Mendiratta [26] in spent hen meat treated with different concentrations of ginger extract, and by Abdel-Naeem and Mohamed [24] in camel-meat burger patties with ginger extract. According to Pawar et al. [27], the decrease in the shear force of meat samples treated with ginger can be explained by the activity of proteolytic enzymes in ginger, as well as by moisture retention and increased water-holding capacity of meat product due to ginger addition. In our study no significant differences were observed in water activity of investigated samples and in moisture content between C and G2 samples, therefore we can assume that the reason for lower hardness of G1 and G2 samples compared to the C sample was, rather, enzyme activity or loosening of the meat matrix by the comminuted-ginger addition. In addition, dissolved collagen derived from connective tissue after ginger treatment has excellent water-binding capacity, and it can improve the tenderness of cooked meat [28].

### 3.5. Sensory Quality

The results of the sensory analysis are shown in Table 5. The addition of ginger and storage time had no significant (*p* > 0.05) effect on the intensity of roasted meat aroma in the tested samples. The scores for this attribute ranged from 6.86 to 7.86 in the control sample, from 6.86 to 7.14 in the sample with 1% ginger addition, and from 6.43 to 7.00 in the sample with 2% ginger addition. The intensity of ginger aroma was moderately perceptible and higher in the sample with 2% (4.29–5.57 points) than 1% ginger addition (3.00–4.57 points). During storage, the intensity of ginger aroma decreased significantly in both samples.

Ginger addition and refrigerated-storage time had no effect (*p* > 0.05) on meatloaf juiciness. The scores for this attribute were in the range of 6.43–7.86 in sample C, 7.00–7.43 in sample G1, and 7.29–7.86 in sample G2. Samples containing ginger were assessed as less cohesive than the control sample only after 21 days of storage, and they were characterised by similar cohesiveness to the control sample on the remaining days of storage.

Ginger-enriched and control meatloaves were characterised by similar softness on storage days 0 and 7. On the remaining days of storage, products containing ginger were evaluated as significantly more tender than the control sample. During storage, the scores for this attribute decreased significantly from 8.86 (day 0) to 7.00 (day 21) in the control sample, but they remained fairly stable in sample G1 (7.43–8.29) and sample G2 (7.71–8.57).

Despite significant differences in texture parameters measured instrumentally, such as hardness and chewiness, between control sample and meatloaves with ginger (Table 4), no adverse effect of ginger addition on the sensorially evaluated texture of products was noted.

The intensity of roasted meat taste and ginger taste was also assessed in the sensory evaluation. The intensity of roasted meat taste was significantly lower in samples with the addition of ginger. This attribute was not affected by storage time, and it received the following scores: 6.86–7.71 in sample C, 5.43–6.57 in sample G1, and 5.86–6.43 in sample G2. Ginger taste was more perceptible in samples with 2% (4.57–5.29 points) than 1% ginger addition (2.71–4.29 points). During storage, changes in the intensity of ginger taste were noted in the meatloaf containing 1% ginger, and this intensity attribute decreased significantly after 14 days of storage.

No significant differences in the overall quality scores were found between the control sample and ginger-enriched samples despite differences in individual attributes. The overall quality scores were 7.43–8.14 in sample C, 8.00–8.71 in sample G1, and 8.00–8.29 in sample G2.

Abdel-Naeem and Mohamed [24] reported an improvement in the juiciness, tenderness, and overall acceptability of camel-meat burgers with the addition of ginger extract. According to Pawar et al. [27], the addition of ginger extract increases the juiciness of meat products by increasing their water-holding capacity. The improvement in the hydrophilic properties of meat products could be attributed to the activity of protease contained in ginger [29]. Abdeldaiem and Ali [25] also observed an improvement in the sensory quality (aroma, flavour, juiciness, and tenderness) of camel meat marinated in fresh ginger extract.

## 4. Conclusions

The present study demonstrated that the addition of ginger rhizome inhibited lipid oxidation during refrigerated storage of pork meatloaves, and oxidative changes were more effectively inhibited in samples with 2% than 1% ginger addition. Meatloaves with the addition of ginger were characterised by higher pH values than the control sample, but these differences were not significant from the technological point of view, and water activity values were similar to those noted in the control sample. Meatloaves with the addition of ginger were also characterised by lower cohesiveness and higher chewiness. In the sensory-quality evaluation, ginger-enriched samples differed from the control sample in aroma, texture, and flavour, but these differences had no significant effect on the overall quality scores. In conclusion, fresh ginger appears to be a superior alternative to synthetic antioxidants in the industrial production of delicatessen meats. Fresh ginger can also be added to home-made meats to obtain products of higher quality and prevent lipid oxidation during storage.

## Figures and Tables

**Table 1 foods-11-03563-t001:** Sensory attributes of pork meatloaves.

Sensory Attributes	Marks of Anchors
**Aroma**	
Roasted meat	Absent—very strong
Ginger	Absent—very strong
**Texture**	
Juiciness	Dry—very juicy
Cohesiveness	Brittle—cohesive
Tenderness	Tough—very tender
**Taste**	
Roasted meat	Absent—very strong
Ginger	Absent—very strong
**Overall quality**	Very poor—excellent

**Table 2 foods-11-03563-t002:** Proximate composition of pork meatloaves and the effect of ginger rhizome and storage time on the pH, water activity, and TBARS values of pork meatloaves.

Parameter	Samples	Storage Time (Days)			
		0	7	14	21
Moisture (%)	C	70.56 ^B^ ± 0.16	ND	ND	ND
	G1	70.17 ^A^ ± 0.13	ND	ND	ND
	G2	70.78 ^B^ ± 0.04	ND	ND	ND
Protein (%)	C	22.45 ^A^ ± 0.45	ND	ND	ND
	G1	22.34 ^A^ ± 0.46	ND	ND	ND
	G2	21.96 ^A^ ± 0.65	ND	ND	ND
Fat (%)	C	6.66 ^A^ ± 0.13	ND	ND	ND
	G1	6.87 ^A^ ± 0.69	ND	ND	ND
	G2	6.75 ^A^ ± 0.24	ND	ND	ND
pH	C	6.13 ^bA^ ± 0.01	6.14 ^bA^ ± 0.01	5.98 ^aA^ ± 0.04	6.13 ^bA^ ± 0.01
	G1	6.18 ^aB^ ± 0.01	6.17 ^aB^ ± 0.02	6.18 ^aB^ ± 0.01	6.19 ^aB^ ± 0.01
	G2	6.14 ^aA^ ± 0.01	6.18 ^bB^ ± 0.01	6.22 ^cC^ ± 0.01	6.22 ^cB^ ± 0.01
Water activity	C	0.977 ^aA^ ± 0.001	0.977 ^aA^ ± 0.002	0.975 ^aA^ ± 0.002	0.975 ^aA^ ± 0.002
	G1	0.977 ^aA^ ± 0.002	0.977 ^aA^ ± 0.003	0.975 ^aA^ ± 0.004	0.975 ^aA^ ± 0.003
	G2	0.977 ^aA^ ± 0.003	0.977 ^aA^ ± 0.004	0.976 ^aA^ ± 0.003	0.975 ^aA^ ± 0.002
TBARS (mg MDA/kg)	C	1.35 ^cC^ ± 0.02	1.07 ^aB^ ± 0.04	1.77 ^dB^ ± 0.05	1.18 ^bC^ ± 0.01
	G1	0.82 ^aB^ ± 0.03	0.78 ^aA^ ± 0.02	1.27 ^bA^ ± 0.25	0.93 ^aB^ ± 0.03
	G2	0.42 ^aA^ ± 0.02	0.82 ^bA^ ± 0.06	1.26 ^cA^ ± 0.01	0.87 ^bA^ ± 0.03

Samples: C—control; G1—1% addition of ginger rhizome; G2—2% addition of ginger rhizome; ND—not determined; ^a–d^—mean values in rows marked with different letters are significantly different at *p* < 0.05; ^A–C^—mean values in columns marked with different letters are significantly different at *p* < 0.05.

**Table 3 foods-11-03563-t003:** The effect of ginger rhizome and storage time on the colour parameters of pork meatloaf.

Parameter	Samples	Storage Time (Days)			
		0	7	14	21
L^*^	C	68.11 ^aA^ ± 1.50	69.45 ^abA^ ± 1,35	70.28 ^bA^ ± 1,03	69.50 ^abA^ ± 0,94
	G1	69.25 ^aA^ ± 1.00	69.51 ^aA^ ± 0.92	70.11 ^aA^ ± 0.59	70.36 ^aAB^ ± 1.17
	G2	69.00 ^aA^ ± 1.16	69.89 ^abA^ ± 1.20	71.22 ^bcB^ ± 0.85	70.77 ^bcB^ ± 1.07
a*	C	8.23 ^aB^ ± 0.63	8.89 ^aB^ ± 0.83	8.31 ^aA^ ± 0.80	10.21 ^bB^ ± 0.69
	G1	7.23 ^aA^ ± 0.43	8.62 ^bAB^ ± 0.24	8.23 ^bA^ ± 0.50	8.58 ^bA^ ± 0.64
	G2	7.73 ^aAB^ ± 0.59	7.97 ^aA^ ± 0.63	8.04 ^aA^ ± 0.60	8.05 ^aA^ ± 0.74
b*	C	13.66 ^bA^ ± 0.82	13.15 ^abA^ ± 0.46	13.54 ^abA^ ± 0.32	12.85 ^aA^ ± 0.61
	G1	13.59 ^aA^ ± 1.17	13.19 ^aA^ ± 0.26	13.13 ^aA^ ± 0.27	13.31 ^aA^ ± 0.56
	G2	13.59 ^bA^ ± 0.40	12.98 ^aA^ ± 0.37	13.24 ^abA^ ± 0.47	13.23 ^abA^ ± 0.50
C*	C	15.96 ^aA^ ± 0.79	15.88 ^aB^ ± 0.76	15.90 ^aA^ ± 0.54	16.43 ^aB^ ± 0.44
	G1	15.41 ^aA^ ± 1.03	15.76 ^aAB^ ± 0.27	15.50 ^aA^ ± 0.28	15.84 ^aAB^ ± 0.68
	G2	15.64 ^aA^ ± 0.58	15.24 ^aA^ ± 0.48	15.50 ^aA^ ± 0.42	15.50 ^aA^ ± 0.43
h°	C	49.45 ^aA^ ± 0.73	50.36 ^aA^ ± 0.68	49.58 ^aA^ ± 0.72	51.98 ^bA^ ± 1.06
	G1	61.87 ^bB^ ± 2.63	56.81 ^aB^ ± 0.83	57.93 ^aB^ ± 1.82	57.21 ^aB^ ± 1.82
	G2	60.40 ^aB^ ± 1.50	58.47 ^aC^ ± 2.04	58.75 ^aB^ ± 2.40	58.69 ^aB^ ± 2.87

Samples: C—control; G1—1% addition of ginger rhizome; G2—2% addition of ginger rhizome; ^a–c^—mean values in rows marked with different letters are significantly different at *p* < 0.05; ^A,B^—mean values in columns marked with different letters are significantly different at *p* < 0.05.

**Table 4 foods-11-03563-t004:** The effect of ginger rhizome and storage time on the texture parameters of pork meatloaf.

Parameter	Samples	Storage Time (Days)			
		0	7	14	21
Hardness (N)	C	12.61 ^bC^ ± 1.20	12.04 ^bB^ ± 1.10	10.57 ^aB^ ± 1.22	10.44 ^aB^ ± 1.69
	G1	10.83 ^bcB^ ± 1.12	11.92 ^cB^ ± 1.21	9.83 ^abB^ ± 1.25	9.04 ^aA^ ± 1.24
	G2	8.32 ^aA^ ± 1.01	9.01 ^aA^ ± 0.82	7,90 ^aA^ ± 1.21	8.81 ^aA^ ± 0.99
Springiness (-)	C	0.81 ^aB^ ± 0.03	0.81 ^aB^ ± 0.04	0.81 ^aB^ ± 0.05	0.82 ^aB^ ± 0.03
	G1	0.78 ^abAB^ ± 0.03	0.80 ^bB^ ± 0.02	0.75 ^aA^ ± 0.04	0.79 ^bAB^ ± 0.04
	G2	0.76 ^abA^ ± 0.04	0.75 ^abA^ ± 0.06	0.72 ^aA^ ± 0.04	0.77 ^bA^ ± 0.03
Cohesiveness (-)	C	0.49 ^aB^ ± 0.04	0.51 ^aB^ ± 0.04	0.53 ^aB^ ± 0.02	0.49 ^aB^ ± 0.03
	G1	0.47 ^aB^ ± 0.03	0.51 ^bB^ ± 0.04	0.46 ^aA^ ± 0.03	0.46 ^aAB^ ± 0.03
	G2	0.43 ^aA^ ± 0.03	0.46 ^aA^ ± 0.03	0.46 ^aA^ ± 0.03	0.44 ^aA^ ± 0.03
Chewiness (J)	C	5.06 ^aC^ ± 0.56	5.04 ^aC^ ± 0.77	4.56 ^aC^ ± 0.98	4.89 ^aB^ ± 0.92
	G1	3.96 ^abB^ ± 0.66	4.16 ^bB^ ± 0.63	3.65 ^aB^ ± 0.73	3.50 ^aA^ ± 0.76
	G2	2.92 ^abA^ ± 0.49	3.51 ^bA^ ± 0.79	2.70 ^aA^ ± 0.56	2.94 ^abA^ ± 0.38

Samples: C—control; G1—1% addition of ginger rhizome; G2—1% addition of ginger rhizome; ^a–c^—mean values in rows marked with different letters are significantly different at *p* < 0.05; ^A–C^—mean values in columns marked with different letters are significantly different at *p* < 0.05.

**Table 5 foods-11-03563-t005:** The effect of ginger rhizome and storage time on the sensory quality of pork meatloaf.

Attributes	Samples	Storage Time (Days)			
		0	7	14	21
Intensity of roasted meat aroma	C	7.86 ^aA^ ± 1.77	8.00 ^aA^ ± 1.41	7.28 ^aA^ ± 1.38	6.86 ^aA^ ± 1.35
1—absent	G1	7.00 ^aA^ ± 0.82	7.14 ^aA^ ± 1.07	6.86 ^aA^ ± 1.07	6.71 ^aA^ ± 0.49
10—very strong	G2	6.86 ^aA^ ± 0.38	7.00 ^aA^ ± 0.82	6.43 ^aA^ ± 0.53	6.43 ^aA^ ± 0.53
Intensity of ginger aroma	C	NE	NE	NE	NE
1—absent	G1	4.57 ^cA^ ± 1.13	4.29 ^bcA^ ± 0.95	3.00 ^aA^ ± 0.58	3.14 ^abA^ ± 0.38
10—very strong	G2	5.43 ^bcA^ ± 0.53	5.57 ^cB^ ± 0.53	4.29 ^aB^ ± 0.76	4.57 ^abB^ ± 0.53
Juiciness	C	7.86 ^aA^ ± 0.69	7.57 ^aA^ ± 0.53	6.57 ^aA^ ± 0.53	6.43 ^aA^ ± 0.79
1—dry	G1	7.43 ^aA^ ± 0.53	7.14 ^aA^ ± 0.90	7.00 ^aA^ ± 0.82	7.43 ^aA^ ± 0.98
10—very juicy	G2	7.86 ^aA^ ± 0.90	7.86 ^aA^ ± 0.69	7.29 ^aA^ ± 0.76	7.29 ^aA^ ± 0.76
Cohesiveness	C	8.71 ^aA^ ± 0.95	8.71 ^aA^ ± 0.95	7.86 ^aA^ ± 0.90	8.29 ^aB^ ± 0.49
1—brittle	G1	8.57 ^bA^ ± 0.53	7.71 ^abA^ ± 0.76	7.14 ^aA^ ± 0.90	7.43 ^aA^ ± 0,79
10—cohesive	G2	8.29 ^aA^ ± 0.95	7.86 ^aA^ ± 1.07	7.29 ^aA^ ± 1.38	7.43 ^aA^ ± 0.53
Tenderness	C	8.86 ^bA^ ± 0.69	8.00 ^abA^ ± 1.00	6.29 ^aA^ ± 0.76	7.00 ^abA^ ± 0.82
1—tough	G1	8.14 ^aA^ ± 1.35	8,00 ^aA^ ± 0.82	7.43 ^aB^ ± 0.98	8.29 ^aB^ ± 0.49
10—very tender	G2	8.57 ^aA^ ± 0.79	8.29 ^aA^ ± 0.49	7.71 ^aB^ ± 0.76	8.29 ^aB^ ± 0.49
Intensity of roasted meat taste	C	7.57 ^aB^ ± 0.79	7.71 ^aB^ ± 0.95	6.86 ^aB^ ± 0.69	7.14 ^aB^ ± 0.69
1—absent	G1	6.57 ^aAB^ ± 0.53	6.57 ^aA^ ± 0.53	5.43 ^aA^ ± 0.53	6.57 ^aAB^ ± 0.79
10—very strong	G2	5.86 ^aA^ ± 0.90	6.43 ^aA^ ± 0.79	6.14 ^aAB^ ± 0.38	5.86 ^aA^ ± 0.69
Intensity of ginger taste	C	NE	NE	NE	NE
1—absent	G1	4.14 ^bA^ ± 0.69	4.29 ^bA^ ± 0.49	2.71 ^aA^ ± 0.76	3.14 ^aA^ ± 0.69
10—very strong	G2	5.29 ^aB^ ± 0.76	5.43 ^aB^ ± 0.76	5.14 ^aB^ ± 0.69	4.57 ^aB^ ± 0.53
Overall quality	C	8.14 ^aA^ ± 1.21	8.14 ^aA^ ± 1.21	7.57 ^aA^ ± 0.98	7.43 ^aA^ ± 0.79
1—very poor	G1	8.71 ^aA^ ± 0.76	8.57 ^aA^ ± 0.79	8.57 ^aA^ ± 0.53	8.00 ^aA^ ± 1.29
10—excellent	G2	8.00 ^aA^ ± 0.82	8.29 ^aA^ ± 0.76	8.14 ^aA^ ± 0.69	8.00 ^aA^ ± 0.82

Samples: C—control; G1—1% addition of ginger rhizome; G2—2% addition of ginger rhizome^: a–c^—mean values in rows marked with different letters differ significantly at *p* < 0.05; ^A,B^—mean values in columns marked with different letters differ significantly at *p* < 0.05; NE—not evaluated.

## Data Availability

Data is contained within the article.

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
