# Peer review of "The Effect of Ginger Rhizome Addition and Storage Time on the Quality of Pork Meatloaf"

_foods, 2022, doi:10.3390/foods11223563_

Round 1
Reviewer 1 Report
The manuscript is very well written :
Line 75: The verb to bake is used in conjunction with baking bread or cakes. In the case of meat, I recommend using the verb to roast.
Chapter 2.2.: I recommend supplementing with a more accurate recipe with regard to the proportions of pork, liquid eggs and salt.
Line 124: The expression 5 mL is redundant, I recommend stating only: Five mL of 0.02 M....
Line 198: In the designation G2, the addition should be 2% ginger and not 1%.
Table 1, table 5 and chapter 3.5.: The term flavor includes both smell and taste. The smell is expressed in the manuscript by the term aroma, in the case of the term flavor I recommend using the term taste.
On lines 338-339 there is a reference to source 27 which explains the effect of ginger on water retention. I recommend stating that in the analyzed samples in the manuscript, no differences were found between the proportions of water in samples C and G2 (Table 2), nor in the water activity values.
In chapter 3.3. I recommend that the authors add an explanation for the difference in L value between the analyzed samples (line 260 onwards).
I recommend mentioning in the discussion that despite the differences in the instrumental texture analysis (Table 4), this did not have a negative impact on the sensory evaluation (Table 5).
Author Response
Responses to the comments contained in the Review I
Line 75: The verb to bake is used in conjunction with baking bread or cakes. In the case of meat, I recommend using the verb to roast.
- According to the recommendation the verb bake has been replaced by the verb roast.
Chapter 2.2.: I recommend supplementing with a more accurate recipe with regard to the proportions of pork, liquid eggs and salt.
- The detailed composition of the pork meatloaves was completed in Sample Preparation section.
Line 124: The expression 5 mL is redundant, I recommend stating only: Five mL of 0.02 M....
- The expression 5 mL has been replaced according to the recommendation.
Line 198: In the designation G2, the addition should be 2% ginger and not 1%.
- This mistake was corrected.
Table 1, table 5 and chapter 3.5.: The term flavor includes both smell and taste. The smell is expressed in the manuscript by the term aroma, in the case of the term flavor I recommend using the term taste.
- The suggested changes of flavor term to taste term was corrected in tables and in the Sensory Quality section.
On lines 338-339 there is a reference to source 27 which explains the effect of ginger on water retention. I recommend stating that in the analyzed samples in the manuscript, no differences were found between the proportions of water in samples C and G2 (Table 2), nor in the water activity values.
- In relation to the comment given by the Reviewer to the lines 338-339, the authors explain that this part of the manuscript is related to the shear force of the samples and presents the view of other authors on the factors which may influence the texture of ginger enriched meat products. Nevertheless, we included suggested information as well as additional explanation to that problem.
In chapter 3.3. I recommend that the authors add an explanation for the difference in L value between the analyzed samples (line 260 onwards).
- The explanation for difference L value between samples was completed in the Colour Parameters section.
I recommend mentioning in the discussion that despite the differences in the instrumental texture analysis (Table 4), this did not have a negative impact on the sensory evaluation (Table 5).
- This information was completed in the Sensory Quality section.
Reviewer 2 Report
Manuscript recieved for review investigated the effect of ginger addion and duration of storage time under refrigitated conditions on the selected quality parameters of pork meat loaf.
Intorduction section is adequate and consise. All needed elements are elaborated. The aim of the study needs corrections.
Material and methodes section is appropriate for the described and conducted testing. More needed corrections are noted in manuscript pdf file.
Main drawback of the applied experimental plan is the lack of microbiological testing, essential for any storage time analysis, esspecially for meat products. Authors have to provide some answers or additional data regading this issue.
Results and discussion section is elaborate and sufficient. Quality of presented results is satisfactory, although, there is a lack of before mentioned microbiological results.
Conclusion section is appropriate for the presented results.
More needed correctoins are noted in manuscripts’ pdf file.
Decission: major revision

Author Response
Response to the comments contained in the Review II
- According to the Reviewer suggestion the aim of the study has been corrected.
- In the Sensory Evaluation section the country where the University of Warmia and Mazury in Olsztyn is located, was added.
- In the present study, microbiological analyses were not performed, because this research is a preliminary study involving indicators of the possibility of ginger rhizome use for this type of product and therefore it relates to the physicochemical evaluation and sensory quality of meatloaf. In further studies, we also plan to expand the evaluation of this type of product to include its microbiological quality.
Round 2
Reviewer 2 Report
The quality of the research is improved, and it is suitable for publication